Prepared for submission to JHEP

# Cosmological cutting rules for Bogoliubov initial states

**Diptimoy Ghosh**[a,1] **Enrico Pajer**[b,2] **Farman Ullah**[a,3]

[a]*Indian Institute of Science Education and Research Pune,*
*Dr. Homi Bhabha Road, Pune 411008, India*

[b]*Department of Applied Mathematics and Theoretical Physics, University of Cambridge*
*Wilberforce Road, Cambridge, CB3 0WA, UK*

*E-mail:* diptimoy.ghosh@iiserpune.ac.in,
farman.ullah@students.iiserpune.ac.in

ABSTRACT: The field theoretic wavefunction in cosmological spacetimes has received much attention as a fundamental object underlying the generation of primordial perturbations in our universe. Assuming an initial Bunch-Davies state, unitary time evolution implies an infinite set of cutting rules for the wavefunction to all orders in perturbation theory, collectively known as the cosmological optical theorem. In this work, we generalise these results to the case of Bogoliubov initial states, accounting for both parity-even and parity-odd interactions. We confirm our findings in a few explicit examples, assuming IR-finite interactions. In these examples, we preserve scale invariance by adiabatically turning on interactions in the infinite past rather than imposing a Bogoliubov state at some finite initial time. Finally, we give a prescription for computing Bogoliubov wavefunction coefficients from the corresponding Bunch-Davies coefficients for both $n$-point contact and four-point exchange diagrams.

## 1 Introduction

Inflation [1–4] is widely accepted as a model for the very early phase of our universe, especially for providing the mechanism of generating primordial perturbations [5, 6]. When the full system is described using quantum mechanics, one expects unitarity to leave an imprint on the observables of the theory. In flat space, unitarity gives rise to the well-known optical theorem for scattering amplitudes and the associated Cutkosky cutting rules [7, 8]. One expects a similar set of relations to exist also in the context of cosmology and this has been confirmed by recent results. In [9–11], an infinite set of equations were derived that relate wavefunction coefficients to all orders in perturbation theory, which are now known as the cosmological optical theorem or cosmological cutting rules. A crucial assumption of these results was an initial Bunch-Davies initial state in the infinite past. A formal extension to more general states was presented in [12]. In this paper *we derive explicit cosmological*

*cutting rules assuming a general Bogoliubov initial state* [13–21]. It should be noted that, in contrast to flat space, the cosmological optical theorem has been formulated so far only perturbatively and research is still in progress to find non-perturbative consequences of unitarity in cosmological spacetimes (see e.g. [22–28]).

The choice of a Bogoliubov initial state is motivated by the desire to see how the cosmological optical theorem generalises to a broader class of initial states, which leads nonetheless to a tractable problem. Keeping this goal in mind, here we consider a class of excited states which are Bogoliubov transformations of the usual Bunch-Davies state. These states are characterised by two rotation-invariant functions of momenta, $(\alpha_k, \beta_k)$, which are known as Bogoliubov coefficients and are constrained by the normalization condition $|\alpha_k|^2 - |\beta_k|^2 = 1$. We assume throughout that these functions are chosen such that the gravitational back-reaction is negligible and the spacetime remains well approximated by de Sitter space. A modified initial state during inflation leaves an imprint on cosmological correlators and this possibility has been constrained by the data [15–20, 29].

The first obstacle in deriving the cutting rules for general Bogoliubov initial states is that the bulk-boundary propagator no longer satisfies the so-called Hermitian analyticity condition, i.e. $K_k^*(\alpha_k, \beta_k) \neq K_{-k}(\alpha_{-k}, \beta_{-k})$, which was the crucial ingredient in the original derivation [9]. However, new sets of relations for the propagator emerge when leveraging the simple analytic dependence on the initial state parameters $\alpha_k$ and $\beta_k$. These new relations, which play the same role as Hermitian analyticity in the Bunch Davies case, form a $\mathbf{Z}_2 \times \mathbf{Z}_2$ discrete symmetry group action on $K_k(\alpha_k, \beta_k)$. Two of these relations can be used to define an appropriate "discontinuity" operation, which in turn leads to two classes of cutting rules. From the new cutting rules we derive constraints for IR-finite $n$-point contact and four-point exchange diagrams. We confirm our results with explicit examples. Throughout the paper, we focus only on massless and conformally coupled scalar fields. We comment briefly on how our cutting rules generalise to massive fields, but we don't present an explicit derivation.

In passing, we were able to derive relations expressing the Bogoliubov wavefunction coefficients in terms of the corresponding Bunch-Davies coefficients for both $n$-point contact and four-point exchange diagrams. In [30], such a relation was derived only for the three-point contact wavefunction coefficient. The relations we derive provide a welcome technical simplification since one does not need to explicitly calculate Bogoliubov wavefunction coefficients from scratch, which would involve performing complicated time integrals over both positive- and negative-frequency modes. Instead, one can just evaluate the Bunch-Davies wavefunction coefficients and use a simple formula to obtain the Bogoliubov equivalent.

The rest of this paper is organized as follows. In Sec. 2 we present a brief review of the cosmological optical theorem and the associated cosmological cutting rules in the case of a Bunch Davies initial state, as derived in [9–11]. In Sec. 3 we derive a new infinite set of cosmological cutting rules for wavefunction coefficients in the case of a Bogoliubov

initial state. In Sec. 4 we discuss the implications of our cutting rules for $n$-point contact and four-point exchange diagrams with IR-finite interactions. In Sec. 6 we give relations between $n$-point and four-point exchange Bogoliubov and the corresponding Bunch-Davies wavefunction coefficients. We conclude in Sec. 7.

### Summary of the main results

For the convenience of the reader, we summarize below our main results:

- We derive a new infinite set of cosmological cutting rules for the wavefunction coefficients $\psi_n$ of the field-theoretic wavefunction, defined in (1.11), assuming Bogoliubov initial states (Eq. (3.6)) and working to all orders in perturbation theory. In this case, we have more than one Hermitian analyticity relation for the bulk-boundary propagator $K$ of the wavefunction,

$$K_{-k}(\alpha_k^*, \beta_k^*) = K_k(\alpha_k, \beta_k)\,, \tag{1.1}$$

$$K_k(\beta_k^*, \alpha_k^*) = K_k(\alpha_k, \beta_k)\,, \tag{1.2}$$

where $\alpha_k, \beta_k$ are the Bogoliubov transformation coefficients. The above relations form a $\mathbf{Z}_2 \times \mathbf{Z}_2$ discrete symmetry group action on $K_k(\alpha_k, \beta_k)$. We use these relations to derive our modified cutting rules given (schematically) as follows,

$$i\mathrm{Disc}(m)\left[i\psi^{(D)}\right]$$

$$= \sum_{\text{cuts}}\left[\prod_{\text{cut momenta}}\int P\right]\prod_{\text{sub-diagram}}(-i)\,\mathrm{Disc}(m)_{\text{internal \& cut lines}}\left[i\psi^{(\text{sub-diagram})}\right]\,, \tag{1.3}$$

where $m = 1, 2$ denotes the two "discontinuity" (Disc) operations defined as

$$\mathrm{Disc}(1)_{\{\alpha_{p_m},\beta_{p_m}\}}\left[f(\{\alpha_{k_i},\beta_{k_i}\},\{\alpha_{p_m},\beta_{p_m}\},\{k_i\},\{\vec{k}_i\},\{p_m\})\right] =$$

$$f(\{\alpha_{k_i},\beta_{k_i}\},\{\alpha_{p_m},\beta_{p_m}\},\{k_i\},\{\vec{k}_i\},\{p_m\}) - f^*(\{\beta_{k_i}^*,\alpha_{k_i}^*\},\{\alpha_{p_m},\beta_{p_m}\},\{k_i\},\{-\vec{k}_i\},\{p_m\})\,, \tag{1.4}$$

$$\mathrm{Disc}(2)_{\{\alpha_{p_m},\beta_{p_m}\}\&\{p_m\}}\left[f(\{\alpha_{k_i},\beta_{k_i}\},\{\alpha_{p_m},\beta_{p_m}\},\{k_i\},\{\vec{k}_i\},\{p_m\})\right] =$$

$$f(\{\alpha_{k_i},\beta_{k_i}\},\{\alpha_{p_m},\beta_{p_m}\},\{k_i\},\{\vec{k}_i\},\{p_m\}) - f^*(\{\alpha_{k_i}^*,\beta_{k_i}^*\},\{\alpha_{p_m},\beta_{p_m}\},\{-k_i\},\{-\vec{k}_i\},\{p_m\})\,, \tag{1.5}$$

where $\{k_i\}$ & $\{p_m\}$ are external and internal energies.

- For the specific case of contact diagrams, these two relations are reduced to the following,

$$\psi_n\left(\{\alpha_{k_i},\beta_{k_i}\},\{k_i\},\{\vec{k}_i\}\right) + \psi_n^*\left(\{\alpha_{k_i}^*,\beta_{k_i}^*\},\{-k_i\},\{-\vec{k}_i\}\right) = 0 \tag{1.6}$$

$$\psi_n\left(\{\alpha_{k_i},\beta_{k_i}\},\{k_i\},\{\vec{k}_i\}\right) + \psi_n^*\left(\{\beta_{k_i}^*,\alpha_{k_i}^*\},\{k_i\},\{-\vec{k}_i\}\right) = 0\,, \tag{1.7}$$

$$. \tag{1.8}$$

Accompanied by scale invariance one can use these relations to derive properties for contact wavefunction coefficients of massless and conformally coupled scalars. We find that for contact massless scalars the $n$-point wavefunction coefficient is anti-symmetric (symmetric) under exchange $\alpha_{k_i} \leftrightarrow \beta_{k_i}$ for parity even (odd) interactions, Eq. (4.7). In the case of conformally coupled scalars for even $n$ and parity even (odd) interaction it is anti-symmetric (symmetric) under the exchange, $\alpha_{k_i} \leftrightarrow \beta_{k_i}$ and for odd $n$ and parity even (odd) interaction, it is symmetric (anti-symmetric), Eq. (4.8). We also find that for both parity odd and even interactions, the $n$-point wavefunction coefficient for massless scalars is Schwarz reflection positive, Eq. (4.9) and for conformally coupled scalars it is Schwarz reflection positive (negative) for even (odd) $n$, Eq. (4.10). We follow similar logic for the exchange cutting rules and derive corresponding relations for the four-point exchange wavefunction coefficients, Eq. (4.14) & Eq. (4.17).

- To confirm the above-mentioned properties, we explicitly compute wavefunction coefficients from Bogoliubov states [13–21, 31–33]. In the literature, these states are often imposed at some finite early time, which leads to an explicit breaking of scale invariance. Here we instead impose these initial conditions in the infinite past and regulate our time integrals by multiplying the interaction Hamiltonian by a factor $e^{\epsilon \tau}$, where $\epsilon > 0$ is taken to zero at the end of the calculation. Physically, this turns off the interactions in the far past bringing us to the free Bogoliubov initial state. For massless scalars, we compute $n$-point contact diagrams using a $\dot{\phi}^n$ interaction, Eq. (5.8) and also compute one four-point exchange diagram with a $\dot{\phi}^3$ interaction, Eq. (5.15). For conformally coupled scalars we compute $n$-point contact diagrams with a $\sigma^n$ interaction, Eq. (5.11). In all cases, our general relations are verified.

- Finally we derive simple algebraic relations expressing Bogoliubov wavefunction coefficients in terms of Bunch-Davies ones. For the case of an $n$-point contact interaction, the result is given by,

$$\psi_n = \sum_{r=0}^{n} \sum_{\sigma} \left[ \prod_{p=1}^{r} \alpha_{k_{\sigma_p}} \prod_{q=r+1}^{n} \beta_{k_{\sigma_q}} \psi_n^{BD} \left( \{k\}_r, -\{k\}_{n-r} \right) \right], \qquad (1.9)$$

where $\{k\}_r = \{k_{\sigma_1}, ...., k_{\sigma_r}\}$ and $\{k\}_{n-r} = \{k_{\sigma_{r+1}}, ...., k_{\sigma_n}\}$ and sum over $\sigma$ runs over all distinct partitions of $(1, 2, 3, \ldots, n)$ into two sets of $r$ and $n-r$ elements respectively. $\psi_n$ is the wavefunction coefficient computed for a Bogoliubov initial state and $\psi_n^{BD}$ is the corresponding Bunch-Davies coefficient. For a four-point exchange diagram the relation can be found in Eq. (6.10).

**Notations and Conventions**

The cosmological background is taken to be de Sitter space with the following metric

$$ds^2 = \frac{-d\tau^2 + d\vec{x}^2}{(H\tau)^2}, \qquad (1.10)$$

where $H$ is the Hubble parameter and $\tau$ is conformal time. Derivatives with respect to conformal time are denoted by a prime, as for example in $\partial_\tau \phi = \phi'$. The field theoretic wavefunction is parameterised as

$$\Psi[\phi, \tau_0] = \exp\left(+\sum_{n=2}^\infty \int \frac{d^3 k_1 \dots d^3 k_n}{n!(2\pi)^{3n}} \psi_n \phi_{\vec{k}_1} \dots \phi_{\vec{k}_n}\right), \qquad (1.11)$$

where $\psi_n = \psi_n(\{\vec{k}\}, \tau_0)$ are the wavefunction coefficients evaluated at time $\tau_0$. The $n$ external momenta are denoted by $\{\vec{k}_a\}$ for $a = 1, 2, \dots, n$ and the $I$ internal energies by $\{p_m\}$ for $m = 1, 2, \dots, I$. We call "energy" the norm of the momenta, $k = |\vec{k}|$. We denote by $\mathcal{K}$ and $\mathcal{G}$ the bulk-boundary and bulk-bulk propagators for a Bunch-Davies initial state, while we use $K$ and $G$ for the case of a Bogoliubov initial state. For useful references on the field-theoretic wavefunction in the cosmological context see [34–41].

## 2 Review of Bunch-Davies Cutting Rules

Here we illustrate the basic idea of the cosmological optical theorem [9] using only the four-point tree-level exchange and $n$-point contact diagrams computed for $\frac{\lambda}{3!}\phi^3$. In [10], these relations were extended to all orders in perturbation theory, including any number of loops and in [11] they were shown to be valid for fields of any mass and spin, for arbitrary interactions, as long as the theory is unitary and the initial state is the Bunch-Davies state. Other constraints from unitarity on the wavefunction were discussed in [42].

The four-point exchange wavefunction Feynman rules give

$$\psi_4(\{k\}, p) = i\lambda^2 \int_{-\infty}^0 d\tau_1 d\tau_2 \mathcal{K}_{k_1}(\tau_1) \mathcal{K}_{k_2}(\tau_1) \mathcal{G}_p(\tau_1, \tau_2) \mathcal{K}_{k_3}(\tau_2) \mathcal{K}_{k_4}(\tau_2), \qquad (2.1)$$

where the wavefunction coefficients $\psi_n$ were defined in (1.11) and $\{k\}$ collectively denotes the four external energies $k_a = |\mathbf{k}_a|$ for $a = 1, \dots, 4$. Using the following property of the Bunch-Davies bulk-boundary propagator,

$$\mathcal{K}_{-k}^*(\tau) = \mathcal{K}_k(\tau), \qquad (2.2)$$

it follows that we can write[1]

$$\psi_4^*(-\{k\}, p) = -i\lambda^2 \int_{-\infty}^0 d\tau_1 d\tau_2 \left[\mathcal{K}_{-k_1}(\tau_1)\mathcal{K}_{-k_2}(\tau_1)\mathcal{G}_p(\tau_1, \tau_2)\mathcal{K}_{-k_3}(\tau_2)\mathcal{K}_{-k_4}(\tau_2)\right]^*,$$
$$= -i\lambda^2 \int_{-\infty}^0 d\tau_1 d\tau_2 \mathcal{K}_{k_1}(\tau_1)\mathcal{K}_{k_2}(\tau_1)\mathcal{G}_p{}^*(\tau_1, \tau_2)\mathcal{K}_{k_3}(\tau_2)\mathcal{K}_{k_4}(\tau_2), \qquad (2.3)$$

Adding Eq. (2.1) and Eq. (2.3), we get,

$$\psi_4(\{k\}, p) + \psi_4^*(-\{k\}, p) = i\lambda \int_{-\infty}^0 d\tau_1 d\tau_2 \mathcal{K}_{k_1}(\tau_1)\mathcal{K}_{k_2}(\tau_1)$$
$$2P_p \text{Im}[\mathcal{K}_p(\tau_1)]\text{Im}[\mathcal{K}_p(\tau_2)]\mathcal{K}_{k_3}(\tau_2)\mathcal{K}_{k_4}(\tau_2), \qquad (2.4)$$

---

[1]Equations in this section are valid for all interactions with an even number of spatial derivatives. In the presence of an odd number of spatial derivatives one needs to simultaneously transform $k_a \to -k_a$ and $\vec{k}_a \to -\vec{k}_a$, which ensures $\partial_{\vec{x}}$ remains invariant.

where we have used the following property,

$$\text{Im}\mathcal{G}_p(\tau_1, \tau_2) = 2P_p\text{Im}[\mathcal{K}_p(\tau_1)]\text{Im}[\mathcal{K}_p(\tau_2)]. \tag{2.5}$$

The key simplification here is that taking the imaginary part of $\mathcal{G}$ removes the Heaviside theta function, which nested the two time integrals. Without the Heaviside theta function, the time integrals are independent and can be recognised to be

$$\psi_4(k_1, k_2, k_3, k_4, p) + \psi_4^*(-k_1, -k_2, -k_3, -k_4, p) = \\ - P_s \left[ \psi_3(k_1, k_2, p) + \psi_3^*(-k_1, -k_2, p) \right] \left[ \psi_3(k_3, k_4, s) + \psi_3^*(-k_3, -k_4, p) \right]. \tag{2.6}$$

Therefore, the cutting rule relates higher point coefficients to lower point ones. Similarly, one can show that for any $n$-point contact diagram, the cutting rule reads,

$$\psi_n(\{k_i\}) + \psi_n^*(\{-k_i\}) = 0. \tag{2.7}$$

This equation has a simple interpretation i.e. there are no internal lines to cut. For parity-odd interactions, we need to be careful about the transformation of the three-momenta. The more precise version of the above formula is

$$\psi_4(\{k_i\}, p, \{\vec{k}_i\}) + \psi_4^*(\{-k_i\}, p, \{-\vec{k}_i\}) = \\ - P_s \left[ \psi_3(\{k_i\}, p, \{\vec{k}_i\}) + \psi_3^*(\{-k_i\}, p, \{-\vec{k}_i\}) \right] \left[ \psi_3(\{k_i\}, p, \{\vec{k}_i\}) + \psi_3^*(\{-k_i\}, p, \{-\vec{k}_i\},) \right], \tag{2.8}$$

and for contact one has,

$$\psi_n(\{k_i\}, \{\vec{k}_i\}) + \psi_n^*(\{-k_i\}, \{-\vec{k}_i\}) = 0. \tag{2.9}$$

## 3  Cutting Rules at any order: Bogoliubov states

In this section, we will derive cutting rules assuming that the initial state is a general Bogoliubov state rather than the Bunch Davies state. Our derivation mimics that presented in [10] and is hence valid to all orders in perturbation theory i.e. for a diagram with any number of vertices, internal lines and loops.

We first briefly review some salient aspects of our setup. The bulk-bulk and bulk-boundary propagators for the Bogoliubov states are denoted by $K_k(\tau)$, $G_s(\tau, \tau')$ and by $\mathcal{K}_k(\tau)$, $\mathcal{G}_s(\tau, \tau')$ for the Bunch-Davies case. The generic expression for the bulk-bulk propagator is given by

$$G_p(\tau_1, \tau_2) = i \left[ \theta(\tau - \tau') \left( \phi_p^+(\tau') \phi_p^-(\tau) - \frac{\phi_p^-(\tau_0)}{\phi_p^+(\tau_0)} \right) \phi_p^+(\tau) \phi_p^+(\tau') + (\tau \leftrightarrow \tau') \right], \tag{3.1}$$

and the bulk-boundary propagator is given by

$$K_k(\tau) = \frac{\phi_k^+(\tau)}{\phi_k^+(\tau_0)}, \tag{3.2}$$

where $\phi_k^+(\tau)$ and $\phi_k^-(\tau)$ are linearly independent solutions of the equation of motion also known as mode functions. For $k \in \mathbb{R}$ we have $\phi^{*+}_k(\tau) = \phi_k^-(\tau)$, which leads to a useful relation between the two propagators,

$$G_p(\tau_1, \tau_2) = iP_p \left[ K^*_{\,p}(\tau_1) K_p(\tau_2)\theta(\tau_1 - \tau_2) + K^*_{\,p}(\tau_2) K_p(\tau_1)\theta(\tau_2 - \tau_1) - K_p(\tau_1)K_p(\tau_2) \right], \tag{3.3}$$

where $P_p$ is the power spectrum evaluated at the late time boundary, i.e

$$P_p = \langle \phi_{\vec{k}}(\tau_0)\phi_{-\vec{k}}(\tau_0) \rangle' = \left| \phi_k^+ \right|^2. \tag{3.4}$$

The prime here means that we have stripped off the momentum conserving delta function. Our conventions for the factors of $i$ in the propagators and the Feynman rules are the same as in [10]. A particular choice for the mode function selects the vacuum state. In particular, given the following Fourier expansion of the field operator $\phi_k$,

$$\phi_k = \phi_k^+ a^\dagger_{-\vec{k}} + \phi_k^- a_{\vec{k}}, \tag{3.5}$$

the state annihilated by the operator $a_{\vec{k}}$ is defined as the vacuum state. Therefore, the most general vacuum state (Bogoluibov state) is given by the following choice of the mode functions

$$\phi_k^+(\tau) = \frac{H}{\sqrt{2k^3}} \left( \alpha_k(1 - ik\tau)e^{ik\tau} + \beta_k(1 + ik\tau)e^{-ik\tau} \right) \qquad \text{(massless scalar)},$$

$$\phi_k^+(\tau) = \frac{H}{\sqrt{2k}} \left( \alpha_k(-i\tau)e^{ik\tau} + \beta_k(i\tau)e^{-ik\tau} \right) \qquad \text{(conformally coupled scalar)}. \tag{3.6}$$

The usual Bunch-Davies vacuum initial state corresponds to the choice of $\alpha_k = 1$, $\beta_k = 0$. For a Bogoliubov state (which is the subject of this paper) $\alpha_k$ and $\beta_k$ can be anything given that they satisfy the constraint equation $|\alpha_k|^2 - |\beta_k|^2 = 1$. This relation imposes the right commutation condition on the creation and annihilation operators i.e. $\left[ a_{\vec{k}}, a^\dagger_{\vec{k}'} \right] = \delta^3(\vec{k}-\vec{k}')$. Although we do not work with any particular choice of the Bogoliubov coefficients we assume that they are chosen such that the backreaction is controlled and the background geometry is well approximated by de Sitter space. The bulk-boundary propagator for a massless scalar with the Bunch-Davies initial condition is then given by

$$\mathcal{K}_k(\tau) = (1 - ik\tau)e^{ik\tau}, \tag{3.7}$$

and for general Bogoliubov states is given by,

$$K(\alpha_k, \beta_k, k) = \frac{1}{(\alpha_k + \beta_k)} \left( \alpha_k(1 - ik\tau)e^{ik\tau} + \beta_k(1 + ik\tau)e^{-ik\tau} \right). \tag{3.8}$$

For the conformally coupled scalar, the bulk-boundary propagator for a Bogoliubov initial state is given by

$$K(\alpha_k, \beta_k, k) = \frac{\tau \left( \alpha_k e^{ik\tau} - \beta_k e^{-ik\tau} \right)}{\tau_0 \left( \alpha_k - \beta_k \right)}, \tag{3.9}$$

where $\tau_0$ is the time on the boundary. Again, the Bunch-Davies case corresponds to $\alpha_k = 1$, $\beta_k = 0$. See [43] for earlier discussions of Bogoliubov states in de Sitter. Notice that the Bogoliubov bulk-boundary propagator does not have a linear in $k$ term when expanded around $k = 0$, just like in Bunch Davies case,

$$\frac{\partial}{\partial k} K(\alpha_k, \beta_k, k)|_{k=0} = 0 \,. \tag{3.10}$$

Here the derivative with respect to $k$ is taken keeping $\alpha_k$ and $\beta_k$ fixed. This simple property leads to the manifestly local test for wavefunction coefficients [44], which must be satisfied for all theories with manifestly local interactions and more generally for soft interactions, as discussed in [45].

Finally, we note that both Bunch-Davies and Bogoliubov bulk-bulk propagators satisfy the following factorization properties,

$$\text{Im}\mathcal{G}(p, \tau_1, \tau_2) = 2P_p\text{Im}[\mathcal{K}(p, \tau_1)]\text{Im}[\mathcal{K}(p, \tau_2)] \,, \tag{3.11}$$

$$\text{Im}G(p, \tau_1, \tau_2) = 2P_p\text{Im}[K(p, \tau_1)]\text{Im}[K(p, \tau_2)] \,. \tag{3.12}$$

## 3.1 Propagator identities

The key ingredient to derive the cosmological optical theorem is the Hermitian analyticity of the Bunch-Davies bulk-boundary propagator [9–11],

$$\mathcal{K}^*_{-k}(\tau) = \mathcal{K}_k(\tau) \,. \tag{3.13}$$

This is then combined with an infinite set of identities for products of the bulk-bulk propagator, the simplest of which is given in (2.5). The bulk-bulk propagator for the Bogoliubov case still has the same factorization properties but now the bulk-boundary propagator satisfies three relations,

$$K^*_{-k}(\alpha^*_k, \beta^*_k) = K_k(\alpha_k, \beta_k) \,, \tag{3.14}$$

$$K^*_k(\beta^*_k, \alpha^*_k) = K_k(\alpha_k, \beta_k) \,, \tag{3.15}$$

$$K_{-k}(\beta_k, \alpha_k) = K_k(\alpha_k, \beta_k) \,. \tag{3.16}$$

This set of transformations actually form a $\mathbf{Z_2} \times \mathbf{Z_2}$ group ($\{a, e\} \times \{b, e\} = \{a, b, ab, e\}$),

$$a\left(K_k\left(\alpha_k, \beta_k\right)\right) = K^*_{-k}\left(\alpha^*_k, \beta^*_k\right) \,, \tag{3.17}$$

$$b\left(K_k\left(\alpha_k, \beta_k\right)\right) = K^*_k\left(\beta^*_k, \alpha^*_k\right) \,, \tag{3.18}$$

$$ab\left(K_k\left(\alpha_k, \beta_k\right)\right) = K_{-k}\left(\beta_k, \alpha_k\right) \,, \tag{3.19}$$

where $a^2 = b^2 = e$. It is important to mention that the above relations are derived by keeping the $k$ dependence of $\alpha_k, \beta_k$ separate from the rest of the $k$ dependence in $K_k(\alpha_k, \beta_k)$. We will assume the same throughout this paper. Note that the first of these relations is the natural extension of Eq. (3.13) to the Bogoliubov case but the other two are new and emerge only because we allow ourselves to vary the parameters $\alpha_k$ and $\beta_k$ appearing in the initial state. This was not possible in the case of a Bunch-Davies state where these parameters are fixed to $\alpha_k = 1$ and $\beta_k = 0$. Now we proceed to prove the cutting rules for the Bogoliubov case to all orders in perturbation theory.

## 3.2 Proof of cutting rules for Bogoliubov initial states

We will follow [10] in proving cutting rules for a general diagram with any number of internal lines and any number of loops. In that work, one first proves a set of "propagator identities" at the level of integrands, which results in an infinite set of identities for the product of propagators. Second, one integrates these relations crucially assuming that coupling constants are real, hence arriving at the final relations among wavefunction coefficients $\psi_n$. However, in our case we do not need to go through this whole procedure. The reason is very simple, the cutting rules proved in [10] rely on the form of the bulk-bulk propagator as a function of the bulk-boundary propagator. This is unchanged in the case of Bogoliubov initial state, (3.3). The only difference is the introduction of two new kinds of "discontinuities" (Discs) because the Bogoliubov bulk-boundary propagator satisfies the new identities mentioned in the preceding section. These are given as

$$
\underset{\{\alpha_{p_m},\beta_{p_m}\}}{\text{Disc}(1)} \left[ f(\{\alpha_{k_i},\beta_{k_i}\},\{\alpha_{p_m},\beta_{p_m}\},\{k_i\},\{\vec{k}_i\},\{p_m\}) \right] =
$$
$$
f(\{\alpha_{k_i},\beta_{k_i}\},\{\alpha_{p_m},\beta_{p_m}\},\{k_i\},\{\vec{k}_i\},\{p_m\}) - f^*(\{\beta_{k_i}^*,\alpha_{k_i}^*\},\{\alpha_{p_m},\beta_{p_m}\},\{k_i\},\{-\vec{k}_i\},\{p_m\}) \,,
$$
$$
\tag{3.20}
$$

$$
\underset{\{\alpha_{p_m},\beta_{p_m}\}\&\{p_m\}}{\text{Disc}(2)} \left[ f(\{\alpha_{k_i},\beta_{k_i}\},\{\alpha_{p_m},\beta_{p_m}\},\{k_i\},\{\vec{k}_i\},\{p_m\}) \right] =
$$
$$
f(\{\alpha_{k_i},\beta_{k_i}\},\{\alpha_{p_m},\beta_{p_m}\},\{k_i\},\{\vec{k}_i\},\{p_m\}) - f^*(\{\alpha_{k_i}^*,\beta_{k_i}^*\},\{\alpha_{p_m},\beta_{p_m}\},\{-k_i\},\{-\vec{k}_i\},\{p_m\}) \,,
$$
$$
\tag{3.21}
$$

where $\{k_i\}$ & $\{p_m\}$ are external and internal energies. These two definitions follow from the propagator relations, (3.14) & (3.15). With all this in mind, we directly start from the propagator lemma mentioned in Eq. (4.7) of [10],

$$
\sum_{\text{cuts; } C \subseteq I}^{2^I} \hat{D}_C = 0 \,,
\tag{3.22}
$$

where $\hat{D}_C$ ("$C$" stands for a "cut" and "$I$" for all internal lines) denotes the imaginary part of the product of internal and "cut" propagators (with appropriate factors of $(2i)$ as given below) present in the "cut" diagram $D_C$. "Cutting" a line with momentum say $\vec{p}$ is defined by replacing the corresponding bulk-bulk propagator $G_p(\tau_1,\tau_2)$ by $-2P_p K_p(\tau_1) K_p(\tau_2)$. The sum above runs over all kinds of cut diagrams $\{D_C\}$, which one can produce from the original diagram $D$. Now, If $D_C$ is disconnected by cut(s) then it is given by the union of connected subdiagrams $D_C^{(n)}$ i.e $D_C = \cup_n D_C^{(n)}$. Mathematically,

$$
\hat{D}_C = \prod_n \text{Im} \left[ (2i)^{L_n} \hat{D}_C^{(n)} \right] \,,
\tag{3.23}
$$

where $L_n$ is the number of loops in the connceted subdiagram $D_C^{(n)}$. One can then write $\hat{D}_C$ in terms of the Discs defined above, (3.20) & (3.21). Let us illustrate this for the simplest case where none of the internal lines are cut, the corresponding term is denoted

by $\hat{D}_{\{\}}$. Now, modulo the external lines, the integrand for the wavefunction coefficient for diagram $D$ reads as follows,

$$\hat{\psi}^{(D)} = i^{1-L} \left[ \prod_{i=1}^{I} \int d^3 p_i G_{p_i} \right].$$ (3.24)

Since the Disc operation does not change the $\alpha_{p_m}$, $\beta_{p_m}$ and $p_m$ dependence of the internal lines, it simply takes the imaginary part of this expression, which in turn is related to $\hat{D}_{\{\}}$ by the above propagator identity,

$$-i\text{Disc}(m)\left[i\hat{\psi}^{(D)}\right] = \left[ \prod_{i=1}^{I} \int d^3 p_i \right] (-2)^{1-L} \hat{D}_{\{\}},$$ (3.25)

where $m$ can be either 1 or 2. Similarly, for diagrams where some internal line(s) are cut, $\hat{\psi}^{D_C}$ is given by the same expression as above except the cut line(s) propagator(s), $G(\tau_1, \tau_2)$ are replaced by $-2PK(\tau_1)K(\tau_2)$. For all cut diagrams, the Disc of $\hat{\psi}^{D_C}$ can be related to $\hat{D}_C$ and we get the following expression,[2]

$$\int d^3 p_1 ... d^3 p_I (-2)^{1-L} \sum_{C \subseteq I}^{2^I} \hat{D}_C = \sum_{C \subseteq I}^{2^I} \left[ \prod_{a \in C} \int d^3 q_a d^3 q'_a P_{q_a q'_a} \right]$$
$$\times \prod_n (-i)\text{Disc}(m) \underset{I_n\{q_a\}}{\left[ i\hat{\psi}^{(D_C^{(n)})} \right]},$$ (3.26)

where $I_n$ ($I_n \subseteq I$) are internal lines contained within $D_C^{(n)}$. Now, the LHS of the above expression vanishes from the propagator lemma, (3.22). Therefore, we have

$$\sum_{C \subseteq I}^{2^I} \left[ \prod_{a \in C} \int d^3 q_a d^3 q'_a P_{q_a q'_a} \right] \prod_n (-i)\text{Disc}(m) \underset{I_n\{q_a\}}{\left[ i\hat{\psi}^{(D_C^{(n)})} \right]} = 0.$$ (3.27)

Finally, we need to multiply this expression by bulk-boundary propagators and integrate over all times. Using propagator identities ((3.14) & (3.15)) for the bulk-boundary propagator, we can move them inside the Disc which proves the cutting rules. Let us show this for Eq. (3.14).

$$\text{Disc}(2)\left[ \prod_a^n K_{k_a}(\alpha_{k_a}, \beta_{k_a})R \right] = \prod_a^n K_{k_a}(\alpha_{k_a}, \beta_{k_a})R - \left( \prod_a^n K_{-k_a}(\alpha_{k_a}^*, \beta_{k_a}^*)R \right)^*,$$ (3.28)

using Eq. (3.14) for any $R$, this becomes,

$$\prod_a^n K_{k_a}(\alpha_{k_a}, \beta_{k_a})\text{Disc}(2)[R].$$ (3.29)

A similar argument holds for $\text{Disc}(1)$. Therefore, the final equation of the cutting rule employing either only $\text{Disc}(1)$ or $\text{Disc}(2)$ is given by

$$i\text{Disc}(m)\left[ i\psi^{(D)} \right] = \sum_{C \subseteq I, C \neq \{\}}^{2^I - 1} \left[ \prod_{a \in C} \int d^3 q_a d^3 q'_a P_{q_a q'_a} \right] \prod_n (-i)\text{Disc}(m) \underset{I_n\{q_a\}}{\left[ i\psi^{(D_C^{(n)})} \right]},$$ (3.30)

---

[2]We refer the reader to the original cutting rules paper for a detailed derivation.

where the cut diagrams on the RHS do not include the "no cut" ($\{\}$) contribution since that has been shifted to the LHS. Also, $\psi$ in the above expressions is the full wavefunction coefficient including bulk-boundary propagators and time integrals.

**Mode-by-mode cutting rules** Above we mainly focus on relations that arise from employing only either Disc(1) or Disc(2) for *all* external propagators. These relations are useful in deriving certain properties of wavefunction coefficients, as we will see in Sec. 4. However, we are not obliged to employ the *same* propagator identity for all external propagators. We could use any one of the two Disc operators for *each* external propagator. This gives us a total of $2^n$ cutting rules for an $n$-point diagram. The additional relations can be very useful if one tries to bootstrap the final answer starting from an ansatz (see [46] for a brief collection of results on the cosmological bootstrap). For example, for a $n$-point contact wavefunction coefficient we have the $2^n$ relations:

$$\psi_n(1, 2, 3, 4, \dots) + \psi_n^*(\bar{1}, \bar{2}, \bar{3}, \bar{4}, \dots) = 0 \,, \tag{3.31}$$

where a bar on the $i$-th external leg denotes either the $a$ or the $b$ operation in (3.17) on the kinematics of that external leg. So for example

$$\bar{1} = (\alpha_{k_1}^*, \beta_{k_1}^*, -k_1 - \vec{k}_1) \quad \text{or} \quad \bar{1} = (\beta_{k_1}^*, \alpha_{k_1}^*, k_1, -\vec{k}_1) \,, \tag{3.32}$$

and so on. We do not explore this any further in this paper.

## 4 Implications of the Cutting Rules for IR-finite interactions

In this section we derive some general implications of our cutting rules, focusing for concreteness on contact and exchange diagrams. Throughout the discussion, we assume IR-finite interactions, so that the scaling of $\psi_n$ fully fixes the scaling with $k$ (i.e. there are no $\eta_0$ regulators).

### 4.1 Contact

We have derived the following two relations for contact wavefunction coefficients,

$$\psi_n\left(\{\alpha_{k_i}, \beta_{k_i}\}, \{k_i\}, \{\vec{k}_i\}\right) + \psi_n^*\left(\{\alpha_{k_i}^*, \beta_{k_i}^*\}, \{-k_i\}, \{-\vec{k}_i\}\right) = 0 \,, \tag{4.1}$$

$$\psi_n\left(\{\alpha_{k_i}, \beta_{k_i}\}, \{k_i\}, \{\vec{k}_i\}\right) + \psi_n^*\left(\{\beta_{k_i}^*, \alpha_{k_i}^*\}, \{k_i\}, \{-\vec{k}_i\}\right) = 0 \,. \tag{4.2}$$

Substituting[3] $k_i \to -k_i$ and $\alpha_{k_i} \to \beta_{k_i}$ in the second relation and using the first, gives another relation,

$$\psi_n\left(\{\alpha_{k_i}, \beta_{k_i}\}, \{k_i\}, \{\vec{k}_i\}\right) = \psi_n\left(\{\beta_{k_i}, \alpha_{k_i}\}, \{-k_i\}, \{\vec{k}_i\}\right) \,. \tag{4.3}$$

Using scale invariance we know,

$$\psi_n\left(\{\alpha_{k_i}, \beta_{k_i}\}, \{k_i\}, \{\vec{k}_i\}\right) = k_T^{3(1-n)+\sum \Delta} f_n\left(\{\alpha_{k_i}, \beta_{k_i}\}, \{k_i/k_T\}, \{\vec{k}_i/k_T\}\right) \,, \tag{4.4}$$

---

[3] As we mentioned early, when we take $k_i \to -k_i$ we leave the $k_i$ dependence in $\alpha$ and $\beta$ unchanged. In other words, we are thinking of $\psi_n$ as a function of independent variables $\alpha$, $\beta$ and $k_i$.

where $\Delta$ is the scaling dimension for the fields in question. For scalars it is defined by,

$$\Delta = \frac{3}{2} + \sqrt{\frac{9}{4} - \frac{m^2}{H^2}}, \qquad (4.5)$$

where $m$ is the mass of the field. Clearly $\Delta = 3$ for massless scalars and $\Delta = 2$ for conformally coupled scalars, which have $m^2 = 2H^2$.

Combining Eq. (4.4) with Eq. (4.3), we get,

$$f_n\left(\{\alpha_{k_i}, \beta_{k_i}\}, \{k_i/k_T\}, \{\vec{k}_i/k_T\}\right) - (-1)^{3(1-n)+\sum \Delta} f_n\left(\{\beta_{k_i}, \alpha_{k_i}\}, \{k_i/k_T\}, \{\vec{k}_i/k_T\}\right) = 0. \qquad (4.6)$$

Two more properties are important. First, for an interaction with $N_s$ spatial derivatives we must have the scaling $\psi_n(-\vec{k}_i) = (-1)^{N_s}\psi_n(\vec{k}_i)$.. Second, for massless fields, $3(1 - n) + \sum \Delta = 3$. Therefore, for massless scalars, combining these observations with (4.6) leads to

$$\psi_n\left(\{\alpha_{k_i}, \beta_{k_i}\}, \{k_i\}, \{\vec{k}_i\}\right) = -(-1)^{N_s}\psi_n\left(\{\beta_{k_i}, \alpha_{k_i}\}, \{k_i\}, \{\vec{k}_i\}\right) \qquad \text{(massless scalar)}, \qquad (4.7)$$

From this, we conclude that *contact wavefunction coefficients for massless fields are anti-symmetric (symmetric) under exchange $\alpha_{k_i} \leftrightarrow \beta_{k_i}$ for parity even (odd) interactions.*

For conformally coupled fields, $3(1 - n) + \sum \Delta = 3 - n$ and therefore,

$$\psi_n\left(\{\alpha_{k_i}, \beta_{k_i}\}, \{k_i\}, \{\vec{k}_i\}\right) = -(-1)^{N_s-n}\psi_n\left(\{\beta_{k_i}, \alpha_{k_i}\}, \{k_i\}, \{\vec{k}_i\}\right) \quad \text{(conformally coupled)}. \qquad (4.8)$$

From this we conclude that *for an even number $n$ of conformally coupled scalars and parity even (odd) interaction, $\psi_n$ is anti-symmetric (symmetric) under the exchange $\alpha_{k_i} \leftrightarrow \beta_{k_i}$. For odd $n$ and parity even (odd) interaction, $\psi_n$ is symmetric (anti-symmetric).*

Scale invariance and Eq. (4.1) imply one more relation for the contact wavefunction coefficient,

$$\psi_n\left(\{\alpha_{k_i}, \beta_{k_i}\}, \{k_i\}, \{\vec{k}_i\}\right) = \psi_n^*\left(\{\alpha_{k_i}^*, \beta_{k_i}^*\}, \{k_i\}, \{\vec{k}_i\}\right) \qquad \text{(massless scalar)}, \qquad (4.9)$$

which is a Schwarz reflection property and,

$$\psi_n\left(\{\alpha_{k_i}, \beta_{k_i}\}, \{k_i\}, \{\vec{k}_i\}\right) = \psi_n^*\left(\{\alpha_{k_i}^*, \beta_{k_i}^*\}, \{k_i\}, \{\vec{k}_i\}\right) \qquad \text{(c.c., } n = \text{even)},$$
$$\psi_n\left(\{\alpha_{k_i}, \beta_{k_i}\}, \{k_i\}, \{\vec{k}_i\}\right) = -\psi_n^*\left(\{\alpha_{k_i}^*, \beta_{k_i}^*\}, \{k_i\}, \{\vec{k}_i\}\right) \quad \text{(c.c., } n = \text{odd)}. \qquad (4.10)$$

These relations we dub Schwarz reflection positive and Schwarz reflection negative. These properties hold for both parity even and parity odd interactions. The relation for massless fields in (4.9) is similar to the corresponding relation for Bunch-Davies initial states but crucially does *not* imply that $\psi_n$ is real. In particular, any real function of $\alpha$ and $\beta$ still satisfies (4.9), but $\psi_n$ is in general complex for complex $\alpha$ and $\beta$. This means that the no-go theorem for the absence of parity-odd interactions at tree level[4] for tensors [48, 49]

---

[4]Parity-odd loop contributions are non-vanishing [47].

and scalars [50–52] do not apply. Indeed this was noticed in [53] where a new shape of parity-odd graviton non-Gaussianity was computed assuming an "$\alpha$-vacuum" initial state, which is a special case of a Bogoliubov transformation. In the special case of real $\alpha$ and $\beta$, the no-go theorems still apply.

## 4.2 Exchange

Here we derive properties for four-point exchange wavefunction coefficient for massless and conformally coupled scalars. The cutting rules for a 4-point exchange diagram read

$$
\begin{aligned}
\psi_4 & \left(\{\alpha_{k_i}, \beta_{k_i}\}, \alpha_p, \beta_p, \{k_i\}, \{\vec{k}_i\}, p\right) + \psi_4^* \left(\{\alpha_{k_i}^*, \beta_{k_i}^*\}, \alpha_p, \beta_p, \{-k_i\}, \{-\vec{k}_i\}, p\right) \qquad (4.11)\\
& = -P_p \left[\psi_3 \left(\{\alpha_{k_i}, \beta_{k_i}\}, \alpha_p, \beta_p, k_1, k_2, \{\vec{k}\}, p\right) + \psi_3^* \left(\{\alpha_{k_i}^*, \beta_{k_i}^*\}, \alpha_p, \beta_p, -k_1, -k_2, \{-\vec{k}\}, p\right)\right]\\
& \qquad \times \left[\psi_3 \left(\{\alpha_{k_i}, \beta_{k_i}\}, \alpha_p, \beta_p, k_3, k_4, \{\vec{k}\}, p\right) + \psi_3^* \left(\{\alpha_{k_i}^*, \beta_{k_i}^*\}, \alpha_p, \beta_p, -k_3, -k_4, \{-\vec{k}\}, p\right)\right],
\end{aligned}
$$

and,

$$
\begin{aligned}
\psi_4 & \left(\{\alpha_{k_i}, \beta_{k_i}\}, \alpha_p, \beta_p, \{k_i\}, \{\vec{k}_i\}, p\right) + \psi_4^* \left(\{\beta_{k_i}^*, \alpha_{k_i}^*\}, \alpha_p, \beta_p, \{k_i\}, \{-\vec{k}_i\}, p\right) \qquad (4.12)\\
& = -P_p \left[\psi_3 \left(\{\alpha_{k_i}, \beta_{k_i}\}, \alpha_p, \beta_p, k_1, k_2, \{\vec{k}\}, p\right) + \psi_3^* \left(\{\beta_{k_i}^*, \alpha_{k_i}^*\}, \alpha_p, \beta_p, k_1, k_2, \{-\vec{k}\}, p\right)\right]\\
& \times \left[\psi_3 \left(\{\alpha_{k_i}, \beta_{k_i}\}, \alpha_p, \beta_p, k_3, k_4, \{\vec{k}\}, p\right) + \psi_3^* \left(\{\beta_{k_i}^*, \alpha_{k_i}^*\}, \alpha_p, \beta_p, k_3, k_4, \{-\vec{k}\}, p\right)\right].
\end{aligned}
$$

The right-hand side of both equations is identical since one can use,

$$
K^*(\beta_k^*, \alpha_k^*, k) = K^*(\alpha_k^*, \beta_k^*, -k). \qquad (4.13)
$$

Therefore,

$$
\begin{aligned}
\psi_4 \left(\{\alpha_{k_i}, \beta_{k_i}\}, \alpha_p, \beta_p, \{k_i\}, \{\vec{k}_i\}, p\right) &= \psi_4 \left(\{\beta_{k_i}, \alpha_{k_i}\}, \alpha_p, \beta_p, \{-k_i\}, \{\vec{k}_i\}, p\right)\\
&= (-1)^{N_s} \psi_4 \left(\{\beta_{k_i}, \alpha_{k_i}\}, \alpha_p, \beta_p, \{-k_i\}, \{-\vec{k}_i\}, p\right)\\
&= -(-1)^{N_s} \psi_4 \left(\{\beta_{k_i}, \alpha_{k_i}\}, \beta_p, \alpha_p, \{-k_i\}, \{-\vec{k}_i\}, -p\right)\\
&= -(-1)^{3(1-n)+\sum \Delta + N_s} \psi_4 \left(\{\beta_{k_i}, \alpha_{k_i}\}, \beta_p, \alpha_p, \{k_i\}, \{\vec{k}_i\}, p\right).
\end{aligned}
$$
$$(4.14)$$

In the above expression while going from the second to the third line we have used the following property of the bulk-bulk propagator,

$$
G_{-p}(\beta_p, \alpha_p, \tau_1, \tau_2) = -G_p(\alpha_p, \beta_p, \tau_1, \tau_2). \qquad (4.15)
$$

This can be derived very easily by combining (3.3) & (3.16). If a time derivative is acting on the internal line then in such a case Feynman rules tell us that we ought to replace (suppressing Bogoliubov arguments) $G_p(\tau_1, \tau_2) \rightarrow \partial_{\tau_1} \partial_{\tau_2} G_p(\tau_1, \tau_2)$ and this produces an

additional term shown as follows,

$$\partial_{\tau_1}\partial_{\tau_2}G_p(\tau_1,\tau_2) = iP_p \left[ K'^*_p(\tau_1)K'_p(\tau_2)\theta(\tau_1-\tau_2) + K'^*_p(\tau_2)K'_p(\tau_1)\theta(\tau_2-\tau_1) - K'_p(\tau_1)K'_p(\tau_2) \right.$$
$$\left. + \underbrace{2i\text{Im}\left[K^*_p(\tau_1)K'_p(\tau_2)\right]\delta(\tau_1-\tau_2)}_{\text{additional term}} \right] .$$
$$(4.16)$$

Clearly $\partial_{\tau_1}\partial_{\tau_2}G_{-p}(\beta_p,\alpha_p,\tau_1,\tau_2) = -\partial_{\tau_1}\partial_{\tau_2}G_p(\alpha_p,\beta_p,\tau_1,\tau_2)$. Therefore, (4.14) implies that the 4-point exchange wavefunction for a massless scalar (and even conformally coupled scalars) is symmetric (anti-symmetric) under exchange $\alpha_{k_i,p} \longleftrightarrow \beta_{k_i,p}$ for a parity even (odd) interaction and anti-symmetric (symmetric) for an odd number of conformally coupled fields. We can prove one more property for the exchange wavefunction as follows,

$$\psi_4\left(\{\alpha_{k_i},\beta_{k_i}\},\alpha_p,\beta_p,\{k_i\},\{\vec{k}_i\},p\right) = -\psi_4^*\left(\{\alpha_{k_i}^*,\beta_{k_i}^*\},\alpha_p^*,\beta_p^*,\{-k_i\},\{-\vec{k}_i\},-p\right)$$
$$= -(-1)^{3(1-n)+\sum\Delta}\psi_4^*\left(\{\alpha_{k_i}^*,\beta_{k_i}^*\},\alpha_p^*,\beta_p^*,\{k_i\},\{\vec{k}_i\},p\right) .$$
$$(4.17)$$

Here, we have used the following property of the bulk-bulk propagator,

$$G_{-p}^*\left(\alpha_p^*,\beta_p^*,\tau_1,\tau_2\right) = G_p\left(\alpha_p,\beta_p,\tau_1,\tau_2\right) \tag{4.18}$$

This property also holds for (4.16) and so (4.17) also is valid in this well. Therefore, the (anti-)Schwarz reflection properties of contact wavefunctions derived above remain at the four-point level. Again the no-go theorems mentioned above do not apply except when all the Bogoliubov coefficients are real.

## 4.3 Generalisation for the massive case

The mode functions for massive scalars are Hankel functions, which have non-trivial analytic structures. In particular, they have a branch point at the origin and a typical choice for the branch cut is along the negative real axis. Therefore, extending the cutting rules and the propagator identities to these mode functions is not straightforward. Especially, those that involve analytically continuing $k_i \to -k_i$ since due to the branch cut, the answer depends on the direction one approaches the negative-real axis. Despite these complications, the cutting rules were proven to hold for the massive case too [9, 11]. In the Bunch-Davies case, one continues the mode functions to the lower-half complex plane and approaches the negative energies from below. This is the natural choice since in this case, all energies have a small imaginary part, $\pm|k_i| - i\epsilon$, $\epsilon > 0$, which comes from the $i\epsilon$ prescription used to project on the interacting vacuum. Now, in our case, we have cutting rules with two kinds of Discs, Eq. (3.20) and Eq. (3.21). Let us first discuss the generalisation of Eq. (3.20) since in this case, we do not analytically continue any energies and therefore the extension is simpler.

The positive and negative frequency mode functions for the massive scalar (Bogoliubov case) are given by,

$$\phi_k^+(\alpha_k, \beta_k, \tau) = \alpha_k \varphi_k^+(\tau) + \beta_k \varphi_k^-(\tau) \,, \tag{4.19}$$

$$\phi_k^-(\alpha_k, \beta_k, \tau) = \alpha_k^* \varphi_k^-(\tau) + \beta_k^* \varphi_k^+(\tau) \,, \tag{4.20}$$

where,

$$\varphi_k^+(\tau) = i e^{-i\frac{\pi}{2}(\nu+\frac{1}{2})} \sqrt{\pi} \frac{H}{2} (-\tau)^{\frac{3}{2}} H_\nu^{(2)}(-k\tau) \,, \tag{4.21}$$

$$\varphi_k^-(\tau) = -i e^{i\frac{\pi}{2}(\nu+\frac{1}{2})} \sqrt{\pi} \frac{H}{2} (-\tau)^{\frac{3}{2}} H_\nu^{(1)}(-k\tau) \,. \tag{4.22}$$

Hankel functions satisfy some useful properties,

$$H_\nu^{(1)*}(z^*) = H_\nu^{(2)}(z) \,, \tag{4.23}$$

$$H_{-\nu}^{(1)}(z) = e^{i\pi\nu} H_\nu^{(1)}(z) \,, \tag{4.24}$$

$$H_{-\nu}^{(2)}(z) = e^{-i\pi\nu} H_\nu^{(2)}(z) \,. \tag{4.25}$$

Now, the bulk-boundary propagator is given by,

$$K_k(\alpha_k, \beta_k) = \frac{\phi^+(\alpha_k, \beta_k, \tau)}{\phi^+(\alpha_k, \beta_k, \tau_0)} \,. \tag{4.26}$$

Using the above formulae (note that $\nu = \sqrt{\frac{9}{4} - \frac{m^2}{H^2}}$ is either real or pure imaginary), one can see,

$$\phi_k^+(\alpha_k, \beta_k, \tau) = \phi_{\,k}^{+\,*}(\beta_k^*, \alpha_k^*, \tau) \,, \tag{4.27}$$

and therefore, we have the usual formula for the bulk-boundary propagator,

$$K_k(\alpha_k, \beta_k, \tau) = K_k^*(\beta_k^*, \alpha_k^*, \tau) \,, \tag{4.28}$$

and since for real $k$, $\phi_k^+(\alpha_k, \beta_k, \tau) = \phi_{\,k}^{-\,*}(\alpha_k, \beta_k, \tau)$, the bulk-bulk propagator expression is same as Eq. (3.3) and we also have,

$$G_p(\alpha_p, \beta_p, \tau_1, \tau_2) = G_p^*(\beta_p^*, \alpha_p^*, \tau_1, \tau_2) \,. \tag{4.29}$$

Therefore, the cutting rule with Disc(1) can be generalised to massive scalars. The generalisation of the cutting rule with Disc(2) is more involved since now for our case there is no preferred analytical continuation because we do not employ an $i\epsilon$ prescription, instead, we turn off interaction by explicitly using an adiabatic function $e^{\epsilon\tau}$ in the Hamiltonian. Also, one should note that the asymptotic limit of bulk-boundary now has both positive and negative exponentials, therefore, in the complex plane one has,

$$\lim_{\tau\to-\infty} K_k(\alpha_k, \beta_k, \tau) \sim e^{i\mathrm{Re}(k)} e^{-\mathrm{Im}(k)\tau} + e^{-i\mathrm{Re}(k)} e^{\mathrm{Im}(k)\tau} \,, \qquad \text{where } k \in \mathbb{C} \,. \tag{4.30}$$

Naively, one might think that the integrals therefore will not converge anywhere in the complex plane but note that we also have an $e^{\epsilon\tau}$ which improves convergence. Therefore, one can perform the integrals assuming that $\epsilon > |\mathrm{Im}(k)|$ and then take $k$ to be real and finally take the limit $\epsilon \to 0$. We leave such explicit calculations for the future.

## 5    Explicit examples

In this section, we perform some explicit checks of the above derived relations. For simplicity, we only take parity even interactions.

### 5.1    On the convergence of time integrals

Before discussing concreted examples, we need to address the issue of convergence of time integrals for Bogoliubov initial states[5]. In the Bunch-Davies case, convergence in the far past is ensured by the prescription $-\infty \to -\infty(1 - i\epsilon)$, which projects the Fock vacuum on the interacting vacuum. In the case of Bogoliubov states, there are a few distinct possibilities. Often one imposes the initial condition at a *finite* initial time $\tau_0 > -\infty$ (see e.g. [17]). This introduces an early time cutoff in the integral. This approach is usually justified by noting that the modes blueshift in the past and therefore the EFT should break down as $\frac{k}{a(\tau_0)} \geq M$, where $M$ is the cutoff for the EFT and $k$ is a mode of interest. This approach displays at least two characteristic properties. First one assumes a Gaussian state at a fixed moment in time for an interacting theory. Second, the final result depends on the cutoff. For example, we get oscillatory features for a sharp cutoff and non-oscillatory behaviour for a smooth cutoff. In this paper, we instead propose a different physical picture and mathematical prescription to regulate the integral[6]. We have in mind a situation in which a free Bogoliubov initial state is prepared in the infinite past where interactions are switched off and the theory is free. Then, interactions are turned on adiabatically and the state evolves according to the Schrödinger equation in the interacting theory. This picture is implemented mathematically by explicitly multiplying the interaction Hamiltonian by a suitable regulator,

$$H(\tau) \to H_\epsilon = H(\tau)R(\epsilon\tau)\,. \tag{5.1}$$

where $R(0) = 1$ and $R(-\infty) = 0$. One can then choose an appropriate regulator (e.g., $e^{\epsilon\tau}$ with $\epsilon > 0$) which turns off the interactions in the past while making the integrals converge (see e.g. [56], or more recent discussions in the cosmological context in [27, 57]). If the limit $\epsilon \to 0$ exists, we can define the results of the regulated calculation. One might ask whether the final answer depends on the choice of $R$, namely the details of how one turns on and off interactions in the past. In Appendix A we show that, for a suitable class of regulators, the final answer is independent of the choice of the regulator. This property puts this prescription on a firmer footing.

**Folded singularities**    Although using a regulator does help with the convergence in the past, we still are faced with the issue of folded singularities. In particular, $\psi_n$ diverges as

---

[5]In this paper, we only consider IR-finite interaction, so we only have to regulate the possible divergence from the far past ($\tau \to -\infty$).

[6]The same approach was implicitly also taken in [54]. Instead, in [20] a procedure of averaging over $\eta_0$ is mentioned but not developed in detail. In practice, in that reference, the contribution from the early boundary is dropped just like we do here invoking the adiabatic switching off of interactions in the infinite past. Finally, it's interesting to notice that the presence of an environment that induces dissipation regulates the integral in a physical way and no artificial cutoffs are necessary [55].

$k_i + k_j \to k_l$ for $i \neq j \neq l$. We call these *folded* configurations. If one uses a cutoff $M$, irrespectively of whether it is sharp or smooth, the folded limit is proportional to $\frac{M}{H}$, which means that the prediction of the EFT is very sensitive to the choice of $M$ for kinematics close to folded triangles. Our interpretation of this is that we can trust our EFT only for configurations that are sufficiently distant from folded divergences. In other words, the EFT prediction is trustworthy everywhere except for configurations very close to folded triangles. Let us make this more precise.

Very generally, if the EFT expansion has to remain valid, we must demand that the corrections given by various operators decrease as the operator dimension increases[7]. Therefore, the correction given by an operator $\mathcal{O}_\Delta$ of dimension $\Delta$ must be larger than the one given by an operator $\mathcal{O}_{\Delta'}$ of dimension $\Delta' > \Delta$. To be more quantitative, consider for simplicity correlators induced by a single contact interaction. On general grounds, we expect that the $n$-point correlation functions have the following dependence on momenta,

$$
\begin{aligned}
B_{\mathcal{O}_\Delta} &\sim \left(\frac{H}{M}\right)^{\Delta-4} \frac{k^{-3n+\Delta}}{\tilde{k}_i^{\Delta-3}} \, , \\
B_{\mathcal{O}_{\Delta'}} &\sim \left(\frac{H}{M}\right)^{\Delta'-4} \frac{k^{-3n+\Delta'}}{\tilde{k}_i^{\Delta'-3}} \, .
\end{aligned}
\tag{5.2}
$$

where we introduced the folded pole $\tilde{k}_i = k_T - 2k_i$ and we assumed that momenta $k_i$ are all of the same order denoted by $k$. The power of $\tilde{k}_i$, call it $p$, is fixed by the following formula given in [59],

$$
p = 1 + \sum_\alpha (\Delta_\alpha - 4) \, ,
\tag{5.3}
$$

where $\Delta_\alpha$ is the mass dimension of operators used to compute the correlation function (only one for this contact case). For the EFT to remain valid we want to satisfy the inequality $B_{\mathcal{O}_{\Delta'}} \ll B_{\mathcal{O}_\Delta}$ and therefore we obtain the condition

$$
\frac{k}{\tilde{k}_i} < \frac{M}{H} \, .
\tag{5.4}
$$

This roughly quantifies how close one can go to the folded limit ($\tilde{k}_i \to 0$) while remaining within the validity of the EFT.

## 5.2 Contact

Let us check some of these relations explicitly with some simple contact examples. We first compute $n$-point contact wavefunction coefficient for a massless scalar field $\phi$, using the following interaction Hamiltonian,

$$
H_\epsilon^\phi = \frac{\lambda}{n!} \int a^{4-n} \phi'^n e^{\epsilon\tau} d^3x \quad \text{with } n \geq 3 \, ,
\tag{5.5}
$$

---

[7]Here we have in mind a simple one-scale power-counting scheme. Of course, more intricate power counting schemes can and do arise in concrete models. See for example [58] for a discussion in the context of inflationary cosmology.

where $a$ is the scale factor and $\lambda$ is the coupling constant. The standard Feynman rules give

$$\psi_n = i\lambda(-H)^{n-4}\prod_{i=1}^{n}\frac{k_i^2}{\alpha_{k_i}+\beta_{k_i}}\int e^{\epsilon\tau}\tau^{2n-4}\prod_{j=1}^{n}\left(\alpha_{k_j}e^{ik_j\tau}+\beta_{k_j}e^{-ik_j\tau}\right)d\tau\,. \tag{5.6}$$

To integrate the above expression we note,

$$\lim_{\epsilon\to 0}\int_{-\infty}^{0}\tau^m e^{ik\tau}e^{\epsilon\tau}d\tau = \frac{(-1)^m m!}{(ik)^{m+1}}\,. \tag{5.7}$$

Now, Eq. (5.6) simplifies to,

$$\psi_n = \lambda(-1)^n(-H)^{n-4}(2n-4)!\prod_{i=1}^{n}\left(\frac{k_i^2}{\alpha_{k_i}+\beta_{k_i}}\right)$$
$$\times\sum_{m=0}^{n}\sum_{\sigma}\frac{\prod_{j=1}^{m}\alpha_{k_{\sigma_j}}\prod_{l=m+1}^{n}\beta_{k_{\sigma_l}}}{(\sum_{j=1}^{m}k_{\sigma_j}-\sum_{j=m+1}^{n}k_{\sigma_j})^{2(n-2)+1}}\,, \tag{5.8}$$

where the sum over $\sigma$ runs over all distinct partitions of $(1,2,\ldots,n)$ into two subsets of $m$ and $n-m$ elements, such that, after the sum over $m$, there are $2^n$ terms. Clearly, the above expression is antisymmetric under exchange $\alpha_{k_i}\leftrightarrow\beta_{k_i}$ and therefore, Eq. (4.7) is satisfied. This expression is also Schwarz reflection positive in agreement with Eq. (4.9). Now, let us compute an $n$-point contact wavefunction coefficient for a conformally coupled scalar, $\sigma$, using the following interaction Hamiltonian,

$$H_\epsilon^\sigma = \frac{\lambda}{n!}\int a^4\sigma^n e^{\epsilon\tau}d^3x \quad\text{with } n\geq 4\,. \tag{5.9}$$

Notice that here we impose $n\geq 4$ because for $n=3$ this interaction gives an IR-divergence. The wavefunction coefficient is given by

$$\psi_n = \frac{i\lambda}{H^4\tau_0^n\prod_{i=1}^{n}(\alpha_{k_i}-\beta_{k_i})}\int e^{\epsilon\tau}\tau^{n-4}\prod_{j=1}^{n}\left(\alpha_{k_i}e^{ik\tau}-\beta_{k_i}e^{-ik\tau}\right)d\tau\,, \tag{5.10}$$

where $\tau_0$ is a late-time cutoff we introduce to be able to extract the leading term for $\tau_0\to 0$. This expression integrates to,

$$\psi_n = \frac{i^{-n}\lambda(-1)^n(n-4)!}{H^4\tau_0^n\prod_{i=1}^{n}(\alpha_{k_i}-\beta_{k_i})}\left[\sum_{m=0}^{n}\sum_{\sigma}\frac{\prod_{j=1}^{m}\alpha_{k_{\sigma_j}}\prod_{j=m+1}^{n}\beta_{k_{\sigma_j}}-\prod_{j=1}^{m}\beta_{k_{\sigma_j}}\prod_{j=m+1}^{n}\alpha_{k_{\sigma_j}}}{2(\sum_{j=1}^{m}k_{\sigma_j}-\sum_{j=m+1}^{n}k_{\sigma_j})^{n-3}}\right]\,, \tag{5.11}$$

where sum over $\sigma$ runs again over all distinct partitions of $(1,2,3,\ldots,n)$ into two sets of $m$ and $n-m$ elements respectively. The expression in the square brackets is anti-symmetric in $\alpha_{k_i},\beta_{k_i}$ for all $n$ but the prefactor $\frac{1}{\prod_{i=1}^{n}(\alpha_{k_i}-\beta_{k_i})}$ is symmetric for even $n$ and anti-symmetric for odd $n$, agreeing with Eq. (4.8). Also, this expression has an $i^{-n}$ present making it Schwarz reflection positive (negative) for $n=$ even (odd) in agreement with Eq.

(4.10). Let us now discuss what these relations imply for the contact correlation functions, $B_n$,

$$B_n = -2 \left[ \prod_{a=1}^{n} \frac{1}{2\mathrm{Re}\ \psi_2} \right] \mathrm{Re}\ \psi_n \,, \tag{5.12}$$

where, for a massless scalar

$$P(k) = \frac{1}{2\mathrm{Re}\ \psi_2(k)} = \frac{H^2}{2k^3} |\alpha_k + \beta_k|^2 \,. \tag{5.13}$$

The above relation implies that if the wavefunction coefficient is a symmetric (anti-symmetric) function of $\alpha_{k_i}, \beta_{k_i}$ then the correlation function is also symmetric (anti-symmetric). Therefore, $B_n$ is anti-symmetric for a massless scalar or for an even number of conformally coupled scalars, while it is symmetric for an odd number of conformally coupled scalars. Finally, we note that for real $\alpha_{k_i}, \beta_{k_i}$, any correlation function involving an arbitrary number of massless scalars and an odd number of conformally coupled scalars must vanish as was already proved in [9].

### 5.3 Exchange

Here, we compute the four-point exchange wavefunction coefficient of a massless scalar for $\frac{\lambda}{3!}\dot{\phi}^3$ interaction.

$$\psi_4 = \frac{-\lambda^2 (k_1 k_2 k_3 k_4)^2 p}{2 \prod_{i=1}^{4} (\alpha_{k_i} + \beta_{k_i})} \int d\tau_1 d\tau_2 e^{\epsilon(\tau_1 + \tau_2)} \tau_1^2 \tau_2^2 \left[ \prod_{i=1}^{2} \left( \alpha_{k_i} e^{ik_i \tau_1} + \beta_{k_i} e^{-ik_i \tau_1} \right) \right.$$

$$\times \prod_{i=3}^{4} \left( \alpha_{k_j} e^{ik_j \tau_2} + \beta_{k_j} e^{-ik_j \tau_2} \right) \left( \left( \alpha_p^* e^{-ip\tau_1} + \beta_p^* e^{is\tau_1} \right) \left( \alpha_p e^{ip\tau_2} + \beta_p e^{-ip\tau_2} \right) \theta(\tau_1 - \tau_2) \right.$$

$$+ (\tau_1 \leftrightarrow \tau_2) - \frac{(\alpha_p + \beta_p)^*}{(\alpha_p + \beta_p)} \left( \alpha_p e^{ip\tau_1} + \beta_p e^{-ip\tau_1} \right) \left( \alpha_p e^{ip\tau_2} + \beta e^{-ip\tau_2} \right) + \frac{i}{p} |\alpha_p + \beta_p|^2 \delta(\tau_1 - \tau_2) \right) \right] \,, \tag{5.14}$$

which gives,

$$
\begin{aligned}
\frac{-\lambda^2 (k_1 k_2 k_3 k_4)^2 p}{2 \prod_{i=1}^4 (\alpha_{k_i} + \beta_{k_i})} & \Big\{ \Big[ \big( \alpha_{k_1} \alpha_{k_2} \alpha_p^* \{ \alpha_{k_3} \alpha_{k_4} \alpha_p \} + \alpha_i \leftrightarrow \beta_i \big) f(k_1 + k_2 - p, \{ k_3 + k_4 + p \}) \\
& + \big( \alpha_{k_1} \alpha_{k_2} \alpha_p^* \{ \alpha_{k_3} \beta_{k_4} \alpha_p \} + \alpha_i \leftrightarrow \beta_i \big) f(k_1 + k_2 - p, \{ k_3 - k_4 + p \}) \\
& + \big( \alpha_{k_1} \alpha_{k_2} \alpha_p^* \{ \beta_{k_3} \alpha_{k_4} \alpha_p \} + \alpha_i \leftrightarrow \beta_i \big) f(k_1 + k_2 - p, \{ -k_3 + k_4 + p \}) \\
& + \big( \alpha_{k_1} \alpha_{k_2} \alpha_p^* \{ \beta_{k_3} \beta_{k_4} \alpha_p \} + \alpha_i \leftrightarrow \beta_i \big) f(k_1 + k_2 - p, \{ -k_3 - k_4 + p \}) \\
& + \big( \alpha_{k_1} \alpha_{k_2} \alpha_p^* \{ \alpha_{k_3} \alpha_{k_4} \beta_p \} + \alpha_i \leftrightarrow \beta_i \big) f(k_1 + k_2 - p, \{ k_3 + k_4 - p \}) \\
& + \big( \alpha_{k_1} \alpha_{k_2} \alpha_p^* \{ \alpha_{k_3} \beta_{k_4} \beta_p \} + \alpha_i \leftrightarrow \beta_i \big) f(k_1 + k_2 - p, \{ k_3 - k_4 - p \}) \\
& + \big( \alpha_{k_1} \alpha_{k_2} \alpha_p^* \{ \beta_{k_3} \alpha_{k_4} \beta_p \} + \alpha_i \leftrightarrow \beta_i \big) f(k_1 + k_2 - p, \{ -k_3 + k_4 - p \}) \\
& + \big( \alpha_{k_1} \alpha_{k_2} \alpha_p^* \{ \beta_{k_3} \beta_{k_4} \beta_p \} + \alpha_i \leftrightarrow \beta_i \big) f(k_1 + k_2 - p, \{ -k_3 - k_4 - p \}) \\
& + \big( \alpha_{k_1} \beta_{k_2} \alpha_p^* \{ ... \} + \alpha_i \leftrightarrow \beta_i \big) f(k_1 - k_2 - p, \{ ... \}) \leftarrow \text{Eight terms} \\
& + \big( \beta_{k_1} \alpha_{k_2} \alpha_p^* \{ ... \} + \alpha_i \leftrightarrow \beta_i \big) f(-k_1 + k_2 - p, \{ ... \}) \leftarrow \text{Eight terms} \\
& + \big( \beta_{k_1} \beta_{k_2} \alpha_p^* \{ ... \} + \alpha_i \leftrightarrow \beta_i \big) f(-k_1 - k_2 - p, \{ ... \}) \leftarrow \text{Eight terms} \Big] \\
& \hspace{6cm} + [k_1, k_2 \longleftrightarrow k_3, k_4] \\
& - \Big( \frac{\alpha_{k_1} \alpha_{k_2} \alpha_p - \alpha_i \leftrightarrow \beta_i}{(k_1 + k_2 + p)^3} + \frac{\alpha_{k_1} \beta_{k_2} \alpha_p - \alpha_i \leftrightarrow \beta_i}{(k_1 - k_2 + p)^3} + \frac{\beta_{k_1} \alpha_{k_2} \alpha_p - \alpha_i \leftrightarrow \beta_i}{(-k_1 + k_2 + p)^3} \\
& \frac{-\beta_{k_1} \beta_{k_2} \alpha_s + \alpha_i \leftrightarrow \beta_i}{(k_1 + k_2 - p)^3} \Big) \times (k_1, k_2 \longrightarrow k_3, k_4) \times \frac{(\alpha_p + \beta_p)^*}{(\alpha_p + \beta_p)} \\
& + \frac{24 \left( |\alpha_p|^2 - |\beta_p|^2 \right)}{p} \Big( \frac{\alpha_{k_1} \alpha_{k_2} \alpha_{k_3} \alpha_{k_4} - \beta_{k_1} \beta_{k_2} \beta_{k_3} \beta_{k_4}}{(k_1 + k_2 + k_3 + k_4)^5} \\
& + \frac{\alpha_{k_1} \alpha_{k_2} \alpha_{k_3} \beta_{k_4} - \beta_{k_1} \beta_{k_2} \beta_{k_3} \alpha_{k_4}}{(k_1 + k_2 + k_3 - k_4)^5} + \frac{\alpha_{k_1} \alpha_{k_2} \beta_{k_3} \beta_{k_4} - \beta_{k_1} \beta_{k_2} \alpha_{k_3} \alpha_{k_4}}{(k_1 + k_2 - k_3 - k_4)^5} \\
& \hspace{8cm} + \text{perm} \Big) \Big\} .
\end{aligned}
$$
(5.15)

The result turned out to be quite lengthy and therefore several shorthand expressions were used. Whenever we have $k_i \leftrightarrow k_j$ it also applies to subscripts i.e $\alpha_{k_i} \leftrightarrow \beta_{k_j}$. The function $f(a, b)$ is given by,

$$
f(a, b) = \frac{-4(a^2 + 5ab + 10b^2)}{b^3 (a + b)^5} .
$$
(5.16)

Finally, anything within {} is repeated in each set of "Eight terms" indicated above as the first eight terms. As expected (5.15) is in agreement with (4.14) and (4.17). To expose the pole structures, the above result can be simplified further by noting that terms appearing in $[k_1, k_2 \longleftrightarrow k_3, k_4]$ are the same as terms above it except with $f(g_1(k_1, k_2) \pm p, g_2(k_3, k_4) \pm p) \longleftrightarrow f(g_2(k_3, k_4) \pm p, g_1(k_1, k_2) \pm p)$, which in words means interchanging only external energy functions appearing in the argument of $f$ without touching $p$. Therefore, one can pairwise sum these functions and simplify the expression further. This does not simplify terms with opposite sign of $p$ in two arguments, e.g., terms like $f(k_1 + k_2 - p, k_3 + k_4 + p) \sim \frac{1}{(k_3 + k_4 + p)^3 (k_1 + k_2 + k_3 + k_4)^5}$ but it does simplify terms with the

same sign of $p$ in both arguments, in other words, terms that would otherwise give poles of the form $\frac{1}{k_1+k_2+k_3+k_4-2p}$. The pairwise sum of such terms makes these poles disappear and we get a product of three-point poles, e.g.,

$$
\begin{aligned}
\left(\alpha_{k_1}\alpha_{k_2}\alpha_p^*\{\beta_{k_3}\beta_{k_4}\beta_p\} + \alpha_i \leftrightarrow \beta_i\right) & \\
\times \left[f(k_1+k_2-p, -k_3-k_4-p) + f(-k_3-k_4-p, k_1+k_2-p)\right] & \\
= \frac{4}{(k_1+k_2-p)^3\,(k_3+k_4+p)^3}\,. &
\end{aligned}
\tag{5.17}
$$

## 6 From Bunch-Davies to Bogoliubov

Since the Bogoliubov mode functions are just linear combinations of positive and negative frequency solutions of Bunch-Davies ones, it is reasonable to expect that one could write the Bogoliubov answer in terms of linear combinations of the Bunch-Davies answer with various signs of energies flipped and weighted by the product of Bogoliubov coefficients. In this section, we derive such relations for interactions without derivatives. This can be generalised in a straightforward way for derivative interactions.

### 6.1 Contact Prescription

The contact wavefunction coefficient is given by,

$$
\psi_n = \frac{i\lambda}{\prod_{i=1}^n (\alpha_{k_i} + \beta_{k_i})} \int \prod_{i=1}^n K_{k_i}(\alpha_{k_i}, \beta_{k_i}) d\tau\,,
\tag{6.1}
$$

$$
= \frac{i\lambda}{\prod_{i=1}^n (\alpha_{k_i} + \beta_{k_i})} \int \prod_{i=1}^n \left(\alpha_{k_i}\mathcal{K}_{k_i}(\tau) + \beta_{k_i}\mathcal{K}_{k_i}^*(\tau)\right) d\tau\,,
\tag{6.2}
$$

where $\mathcal{K}_k(\tau)$ is the Bunch-Davies bulk-boundary propagator. Using $\mathcal{K}_{-k}(\tau) = \mathcal{K}_k^*(\tau)$, the above equation can be rewritten as,

$$
\psi_n = \frac{i\lambda}{\prod_{i=1}^n (\alpha_{k_i} + \beta_{k_i})} \int \prod_{i=1}^n \left(\alpha_{k_i}\mathcal{K}_{k_i}(\tau) + \beta_{k_i}\mathcal{K}_{-k_i}(\tau)\right) d\tau\,,
\tag{6.3}
$$

which can be expanded as follows,

$$
\psi_n = \frac{i\lambda}{\prod_{i=1}^n (\alpha_{k_i} + \beta_{k_i})} \sum_{m=0}^n \sum_\sigma \left(\prod_{j=1}^m \alpha_{k_{\sigma_j}} \prod_{j=m+1}^n \beta_{k_{\sigma_j}} \int d\tau \prod_{j=1}^m \mathcal{K}_{k_{\sigma_j}} \prod_{k=m+1}^n \mathcal{K}_{-k_{\sigma_j}}\right)\,,
\tag{6.4}
$$

where the sum over $\sigma$ runs over all distinct bipartitions of $(1, 2, \ldots, n)$ into two sets of size $m$ and $n-m$, respectively. The above relation can be expressed in terms of the Bunch-Davies wavefunction coefficients,

$$
\psi_n = \sum_{r=0}^n \sum_\sigma \left[\prod_{p=1}^r \alpha_{k_{\sigma_p}} \prod_{q=r+1}^n \beta_{k_{\sigma_q}} \psi_n^{BD}\left(\{k\}_r, -\{k\}_{n-r}\right)\right]\,,
\tag{6.5}
$$

where $\{k\}_r = \{k_{\sigma_1}, \ldots, k_{\sigma_r}\}$ and $\{k\}_{n-r} = \{k_{\sigma_{r+1}}, \ldots, k_{\sigma_n}\}$ and $\sigma$ runs over all distinct partitions of $(1, 2, 3, \ldots, n)$ into two sets of $r$ and $n-r$ elements respectively.

## 6.2 Exchange prescription

For the contact case, we only needed the expression of Bogoliubov bulk-boundary propagator in terms of Bunch-Davies one but for the four-point exchange case, one needs a similar expression for the bulk-bulk propagator. We find,

$$
G_p(\alpha_p, \beta_p, \tau_1, \tau_2) = |\alpha_p|^2 \mathcal{G}_p(\tau_1, \tau_2) - |\beta_p|^2 \mathcal{G}_{-p}(\tau_1, \tau_2) + A\mathcal{K}_p(\tau_1)\mathcal{K}_p(\tau_2)
$$
$$
+ B\mathcal{K}_{-p}(\tau_1)\mathcal{K}_{-p}(\tau_2) - \frac{(\alpha_p + \beta_p)^* \alpha_p \beta_p}{2p^3 (\alpha_p + \beta_p)} \left[\mathcal{K}_p(\tau_1)\mathcal{K}_{-p}(\tau_2) + \mathcal{K}_{-p}(\tau_1)\mathcal{K}_p(\tau_2)\right] ,
\tag{6.6}
$$

where

$$
A = \frac{1}{2p^3}\left(|\alpha_p|^2 + \alpha_p \beta_p^* - \frac{\alpha_p^2 (\alpha_p + \beta_p)^*}{(\alpha_p + \beta_p)}\right) ,
\tag{6.7}
$$

$$
B = \frac{1}{2p^3}\left(|\beta_p|^2 + \alpha_p^* \beta_p - \frac{\beta_p^2 (\alpha_p + \beta_p)^*}{(\alpha_p + \beta_p)}\right) ,
\tag{6.8}
$$

and $\mathcal{G}_p(\tau_1, \tau_2)$ is the bulk-bulk propagator for the Bunch-Davies case. Let us now compute the four-point exchange diagram for a massless scalar field.

$$
\psi_4 = \frac{-\lambda^2}{\prod_{i=1}^4 (\alpha_{k_i} + \beta_{k_i})} \int d\tau_1 d\tau_2 \prod_{i=1}^2 (\alpha_{k_i}\mathcal{K}_{k_i}(\tau_1) + \beta_{k_i}\mathcal{K}_{-k_i}(\tau_1)) \prod_{i=3}^4 (\alpha_{k_i}\mathcal{K}_{k_i}(\tau_2) + \beta_{k_i}\mathcal{K}_{-k_i}(\tau_2))
$$
$$
\times \left(|\alpha_p|^2 \mathcal{G}_p(\tau_1, \tau_2) - |\beta_p|^2 \mathcal{G}_{-p}(\tau_1, \tau_2) + A\mathcal{K}_p(\tau_1)\mathcal{K}_p(\tau_2) + B\mathcal{K}_{-p}(\tau_1)\mathcal{K}_{-p}(\tau_2)\right.
$$
$$
\left. - \frac{(\alpha_p + \beta_p)^* \alpha_p \beta_p}{2p^3 (\alpha_p + \beta_p)} \left[\mathcal{K}_p(\tau_1)\mathcal{K}_{-p}(\tau_2) + \mathcal{K}_{-p}(\tau_1)\mathcal{K}_p(\tau_2)\right]\right) ,
\tag{6.9}
$$

which simplifies to,

$$
|\alpha_p|^2 \left[\alpha_{k_1}\alpha_{k_2}\{\alpha_{k_3}\alpha_{k_4}\}\psi_4^{BD}(k_1, k_2, \{k_3, k_4\}) + \alpha_{k_1}\alpha_{k_2}\{\alpha_{k_3}\beta_{k_4}\}\psi_4^{BD}(k_1, k_2, \{k_3, -k_4\})\right.
$$
$$
\alpha_{k_1}\alpha_{k_2}\{\beta_{k_3}\alpha_{k_4}\}\psi_4^{BD}(k_1, k_2, \{-k_3, k_4\}) + \alpha_{k_1}\alpha_{k_2}\{\beta_{k_3}\beta_{k_4}\}\psi_4^{BD}(k_1, k_2, \{-k_3, -k_4\})
$$
$$
+ \underbrace{\alpha_{k_1}\beta_{k_2}\{...\}\psi_4^{BD}(k_1, -k_2, \{...\})}_{\text{Four terms}} + \underbrace{\alpha_{k_2}\beta_{k_1}\{...\}\psi_4^{BD}(-k_1, k_2, \{...\})}_{\text{Four terms}}
$$
$$
+ \underbrace{\beta_{k_1}\beta_{k_2}\{...\}\psi_4^{BD}(-k_1, -k_2, \{...\})}_{\text{Four terms}}\right] - |\beta_p|^2 \left[p \longrightarrow -p\right]
$$
$$
+ A\left[\psi_4^{BD}(k_i, k_j, k_m, k_n) \longrightarrow \psi_3^{BD}(k_i, k_j, p)\psi_3^{BD}(k_m, k_n, p)\right]
$$
$$
+ B\left[p \longrightarrow -p, \psi_4^{BD}(k_i, k_j, k_m, k_n) \longrightarrow \psi_3^{BD}(k_i, k_j, p)\psi_3^{BD}(k_m, k_n, p)\right]
$$
$$
- \frac{(\alpha_+ \beta_p)^*}{2p^3 (\alpha_p + \beta_p)} \left[\psi_4^{BD}(k_i, k_j, k_m, k_n) \longrightarrow \psi_3^{BD}(k_i, k_j, p)\psi_3^{BD}(k_m, k_n, -p)\right.
$$
$$
\left. + \psi_4^{BD}(k_i, k_j, k_m, k_n) \longrightarrow \psi_3^{BD}(k_i, k_j, -p)\psi_3^{BD}(k_m, k_n, p)\right] .
\tag{6.10}
$$

Anything within {} is repeated in each set of "Four terms" indicated above in the same order as the first four terms. As one can see the relation is already quite complicated for

four-point exchange and is expected to get even worse for higher-point non-contact diagrams, therefore such a prescription may not be of much practical use for these higher-point non-contact diagrams.

## 7 Conclusions

In this work, we derived cosmological cutting rules for Bogoliubov initial states to all orders in perturbation theory. We derived implications of these cutting rules for $n$-point contact and four-point exchange diagrams and checked these implications explicitly for specific examples of wavefunction coefficients involving massless and conformally coupled scalars. Finally, we gave relations expressing Bogoliubov wavefunction coefficients in terms of Bunch-Davies ones both for $n$-point contact and four-point exchange wavefunction coefficients. Interesting avenues for future work include:

- It would be nice to extend to all orders and to massive fields our relations between the Bogoliubov and Bunch-Davies wavefunction coefficients.

- The generalisation to massless spinning fields should be straightforward since they have the same mode functions as that of massless scalars except for the polarization factors.

- In the future one would also like to study the case of more general initial states, e.g., coherent states. The difficulty in approaching this problem is that a coherent state is not a Bogoliubov transform of the usual Bunch-Davies state.

### Acknowledgments

We acknowledge that this collaboration started during the workshop Correlators in Cortona, which was held in September 2023 in Cortona, Italy. DG acknowledges support from the Core Research Grant CRG/2023/001448 of the Anusandhan National Research Foundation (ANRF) of the Gov. of India. This work has been supported by STFC consolidated grant ST/X001113/1, ST/T000694/1, ST/X000664/1 and EP/V048422/1.

## A Proof of regulator independence

To show regulator independence we follow the derivation given in a physics stack exchange post by Sangchul Lee[8]. Since throughout the paper we only consider IR-finite interactions, the integrals we encounter are of the form,

$$I_\epsilon = \int_{-\infty}^0 \tau^n e^{ik\tau} R(\epsilon\tau)d\tau, \qquad (A.1)$$

---

[8]https://math.stackexchange.com/questions/1968568/regulating-int-0-infty-sin-x-mathrmd-x

where $n \geq 0$ and $R$ is a regulator controlled by a small parameter $\epsilon$. We assume that the regulator satisfies the property, $R(0) = 1$. We are interested in the limit $\epsilon \to 0$ of the regulated expression, $I_0 = \lim_{\epsilon \to 0} I_\epsilon$. Let us substitute $\tau = -\tau'$, we get,

$$= (-1)^n \int_0^\infty \tau^n e^{-ik\tau} \tilde{R}(\epsilon\tau) d\tau, \tag{A.2}$$

where $\tilde{R}(\epsilon\tau) = R(-\epsilon\tau)$. Let us write $\tilde{R}(\epsilon\tau)$ in the following integral form,

$$\tilde{R}(x) = \int_x^\infty \rho(t) dt, \tag{A.3}$$

where $\rho(t)$ is integrable for $t \in [0, \infty]$. This simply means that the derivative of the regulator is integrable. Using these facts we write,

$$
\begin{aligned}
I_0 &= \lim_{\epsilon \to 0} \int_0^\infty \frac{1}{i^n} \frac{d^n}{dk^n} e^{-ik\tau} \tilde{R}(\epsilon\tau) d\tau, \\
&= \lim_{\epsilon \to 0} \frac{1}{i^n} \frac{d^n}{dk^n} \int_0^\infty e^{-ik\tau} \tilde{R}(\epsilon\tau) d\tau, \\
&= \frac{1}{i^n} \frac{d^n}{dk^n} \lim_{\epsilon \to 0} \lim_{a \to \infty} \int_0^a e^{-ik\tau} \tilde{R}(\epsilon\tau) d\tau
\end{aligned}
\tag{A.4}
$$

Since $\left| e^{-ik\tau} \tilde{R}(\epsilon\tau) \right| = \tilde{R}(\epsilon\tau) \; \forall k \in \mathbb{R}$, which is integrable (we have $\epsilon > 0$), we can take the derivative outside the integral. cutting

$$= \frac{1}{i^n} \frac{d^n}{dk^n} \lim_{\epsilon \to 0} \lim_{a \to \infty} \int_0^a e^{-ik\tau} \left( \int_{\epsilon\tau}^\infty \rho(x) dx \right) d\tau \tag{A.5}$$

Now using Fubini's theorem we interchange the order of integration keeping in mind that we originally have $x$ integration from $\epsilon\tau$ to $\infty$ which means $\tau < \frac{x}{\epsilon}$ but notice also, $\tau < a$ (from (A.5)), therefore, $\tau < \min\{a, \frac{x}{\epsilon}\}$,

$$
\begin{aligned}
&= \frac{1}{i^n} \frac{d^n}{dk^n} \lim_{\epsilon \to 0} \lim_{a \to \infty} \int_0^\infty \left( \int_0^{\min\{a, \frac{x}{\epsilon}\}} e^{-ik\tau} d\tau \right) \rho(x) dx, \\
&= \frac{1}{i^n} \frac{d^n}{dk^n} \lim_{\epsilon \to 0} \lim_{a \to \infty} \int_0^\infty \left( \frac{e^{-ik\min\{a, \frac{x}{\epsilon}\}} - 1}{-ik} \right) \rho(x) dx
\end{aligned}
\tag{A.6}
$$

Now, one can use the Dominated convergence theorem to commute the $\lim_{a \to \infty}$ inside the integral

$$\frac{1}{i^n} \frac{d^n}{dk^n} \lim_{\epsilon \to 0} \int_0^\infty \left( \frac{e^{-ik\frac{x}{\epsilon}} - 1}{-ik} \right) \rho(x) dx = \frac{1}{i^n} \frac{d^n}{dk^n} \int_0^\infty \frac{\rho(x)}{ik} dx = \frac{1}{i^n} \frac{d^n}{dk^n} \frac{1}{ik}, \tag{A.7}$$

where to derive the last line we have used the Riemann-Lebesgue lemma and the fact that

$$\int_0^\infty \rho(x) dx = \tilde{R}(0) = R(0) = 1. \tag{A.8}$$

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
