# Peer review of "Cosmological cutting rules for Bogoliubov initial states"

_SciPost Physics_

## Round 1 · Referee Report · Anonymous (Referee 1) · 2024-10-25

Report

The manuscript generalized the known cutting rules of cosmological correlators to theories with non-Bunch-Davies (non-BD) initial states. These cutting rules are derived by an explicit analysis of Green functions with non-BD coefficients. The authors further described how to deal with early time divergences usually encountered in the study of non-BD theories, and showed how to understand the folded limits of non-BD correlators. I think the results are technically valid and useful for understanding non-BD theories in cosmology. I have three small questions for the authors, listed below. With these questions addressed, I'll be happy to recommend the paper for publication at SciPost.

1. The cutting rules derived in this paper relate several correlators with different Bogoliubov coefficients, i.e., different initial conditions. Normally, we think of a theory as fully specified by the Lagrangian and the initial condition. Therefore, correlators with different Bogoliubov coefficients correspond to objects of different theories, rather than different objects in a given theory. This is in contrast to previous cutting rules derived for BD states, where we do have a relation connecting different objects in a given theory. This makes the physical meaning of the non-BD cutting rule a little obscured. I suggest the authors to clarify this point or at least to provide some comments on this issue.

2. It is often said that the cutting rule is a consequence of bulk unitarity, similar to the optical theorem of flat-space amplitudes. On the other hand, the cutting rules in this work are derived by a direct manipulation of propagators instead of a unitarity condition (such as the unitarity of the S matrix). Can authors provide more explicit relations between the bulk unitarity and the cutting rule for non-BD theories?

3. The prescription for removing early time divergences in the paper involves a seemingly time-dependent coupling. Can this time dependence be realized in a more realistic model? Also, does this time-dependent coupling break the scale symmetry of the theory? If not, is there any underlying reason?

Recommendation

Ask for minor revision

  • validity: -
  • significance: -
  • originality: -
  • clarity: -
  • formatting: -
  • grammar: -

Author:  Farman Ullah  on 2024-11-05  [id 4932]

(in reply to Report 1 on 2024-10-25)
Category:
answer to question
reply to objection

We thank the referee for investigating our manuscript and providing useful comments. Below is our response.

Response to query 1:

We agree with the referee that our cutting rule equations
relate some n-point wavefunction coefficient with a particular choice of
Bogoliubov coefficients to the wavefunction coefficients where some of the
coefficients are interchanged or even complex conjugated. This relates
objects in the same theory with different initial conditions. It is indeed
a good idea to add a comment to the paper to clarify this point. As to
the more philosophical point of “Normally, we think of a theory as fully
specified by the Lagrangian and the initial condition” we have perhaps a
different point of view. The standard model is still the standard model
both at LEP, where we collide electrons, and at LHC, where we collide
protons. In other words, our cutting rules are not so different from, say,
soft theorems or crossing relations for amplitudes. A soft theorem relates
A(hard → hard+soft) to A(hard → hard). If I think of all particles in-
going, the first and the second processes have different initial conditions,
but most people would say we are constraining the same theory (say QED
or GR).

Response to query 2:

We thank the referee for raising this important issue. Unitarity
was used in the step going from the propagator identities, which
are valid irrespectively of unitarity (they are just generalizations of θ(x) +
θ(−x) = 1) to the relation among wavefunction coefficients. In this step,
we used that the coupling constants are real so they are not affected by the
complex conjugation. Notice that we are always working with real fields
and therefore, the information of unitarity is encoded in the fact that
couplings are assumed to be real. This was also the same place where
unitarity was used in arXiv:2103.09832. To clarify this point further, we
are eager to add a few comments to the manuscript.

Response to query 3:

The time-dependent adiabatic function is just a mathematical procedure to select the correct state at τ → −∞.
The fictitious time dependence is removed at the end of the calculation where we take ϵ → 0.
The fact that such a time dependence in the Hamiltonian does not affect
the scale-invariance of the final results is clear from the proof discussed in
Appendix A. In other words, the time dependence of the Hamiltonian is
a convenient regulator to specify how one computes an indefinite integral
extending to τ = −∞, where the integrand oscillates. In the appendix, we
show that, when we remove the regulator the final result does not depend
on what regulator we had used.

We thank the referee again for their useful comments. We hope we have addressed
the concerns of the referee and that our manuscript will be seen fit for
publication to Scipost.

Attachment:

scipost_report.pdf

---

## Round 1 · Referee Report · Anonymous (Referee 2) · 2024-11-5

Strengths

1. This paper investigates the cutting rules in a non-trivial initial state where the mode functions no longer behave in a simple way un complex conjugation. The cutting rules provides an interesting perspective on this scenario which has been discussed in the literature.

2. Their analysis is complete and novel.

Weaknesses

1. The preparation of the state in the interacting theory makes sense in the $\epsilon \to 0$ limit, but can be subtle. Appendix A covers some of these issues, but this could be given more attention.

Report

This paper provides a novel perspective on excited initial states in cosmology, through cutting rules. The excited states do not obey the same basic complex conjugation rules as the Bunch-Davies fields and thus this is a non-trivial extension of previous work on cosmological cutting rules. The paper is detailed and gives several non-trivial examples. Perturbation theory in these excited states is somewhat more delicate than the Bunch-Davies case, but the issue of preparing the state (etc) is handled with sufficient care. I recommend for publication in the current form.

Recommendation

Publish (easily meets expectations and criteria for this Journal; among top 50%)

---

## Round 1 · Referee Report · Anonymous (Referee 1) · 2024-11-5

Report

The authors have addressed my questions adequately and I am happy to recommend the current version of the manuscript for publication at SciPost.

Recommendation

Publish (easily meets expectations and criteria for this Journal; among top 50%)

---

## Editorial Decision

resubmitted